# Low Noise Interface ASIC of Micro Gyroscope with Ball-disc Rotor

**DOI:** 10.3390/s20041238

**Published:** 2020-02-24

**Authors:** Mingyuan Ren, Honghai Xu, Xiaowei Han, Changchun Dong, Xuebin Lu

**Affiliations:** 1School of Software and Microelectronics, Harbin University of Science and Technology, Harbin 150080, China; rmy2000@126.com (M.R.); honghai9960605@163.com (H.X.); hitdongcc@163.com (C.D.); lxbsw@126.com (X.L.); 2School of Computer and Information Engineering, Harbin University of Commerce, Harbin 150028, China

**Keywords:** low noise interface, micro gyroscope, capacitive detection

## Abstract

A low noise interface ASIC for micro gyroscope with ball-disc rotor is realized in 0.5µm CMOS technology. The interface circuit utilizes a transimpedance pre-amplifier which reduces input noise. The proposed interface achieves 0.003°/s/Hz^1/2^ noise density and 0.003°/s sensitivity with ±100°/s measure range. The functionality of the full circuit, including circuit analysis, noise analysis and measurement results, has been demonstrated.

## 1. Introduction

Micro gyroscopes are small, inexpensive sensors that can measure angular rate or angle, which have increasingly taken the place of conventional rate sensors in many applications such as defense and consumer electronics [1]. Vibratory gyroscopes are the most common of the existing micro gyroscopes [2]. But it is difficult to reach tactical-grade and inertial-grade performance with these kinds of gyroscopes because of the structural constraint. Rotor gyroscope potentially has a higher performance than vibratory gyroscopes [3,4,5]. 

Noise isn’t only an area of science and technology that poses practical problems but also has deep intellectual attractions [1]. The noise in micro gyroscopes includes both external noise and internal noise. The magnitude of external noise signals coupled into a gyroscope varies with the actual conditions. Mechanical thermal noise and circuit noise are the most important sources of internal noise in micro gyroscopes. Effectively reducing internal noise and improving the signal-to-noise ratio of the system are important challenge for gyroscope designers.

The mechanical noise exists widely in micro gyroscope structures, namely Brownian motion, and is caused by random impacts of molecules on a structure, it is commonly considered as the mechanical sensitivity limitation [6,7,8,9]. And the circuit noise includes electrical-thermal noise (Johnson noise), shot noise, flicker noise (1/f noise), demodulation phase noise, and so on. To reduce this noise, low-noise readout circuits have been reported [10,11,12,13,14,15,16]. But, the published papers mostly focus on vibratory micro gyroscope, and the noise analysis of rotor-type micro gyroscope is rarely mentioned. As the noise level affects the detection accuracy of micro gyroscope, it is urgent to carry out the research on the related theory and model of rotor micro gyroscope noise.

In order to improve the performance and detection accuracy of the rotor micro gyroscope, it is necessary to reduce the noise level of the system. At the same time, noise suppression is the core content of ASIC research on micro gyroscope interface. In this paper, the noise model of ASIC system with ball-disc rotor micro gyroscope interface will be established according to the unique damping characteristics and driving mode of ball-disc rotor micro gyroscope. 

## 2. System Noise of Micro Gyroscope 

The mechanical structure of the ball-disc rotor micro gyroscope is shown in Figure 1. According to the different sources of noise, system noise of the micro gyroscope can be divided into mechanical noise, driving noise and electronic noise.

### 2.1. Mechanical Noise

Considering mechanical noise is produced by the thermal motion of micro gyroscope rotor, the thermal motion equation of the micro gyroscope can be described as:(1)$$I\ddot{\alpha }+D\dot{\alpha }+E\alpha =F,$$
where *F* denotes the torque of the random thermal motion; I denotes the rotor moment of inertia; *D* denotes damping coefficient; *E* denotes the elastic coefficient, radial component coefficient of electromagnetic force.

When the system reaches thermal equilibrium, the spectral density of random wave moment of Brownian thermal motion is [11]:(2)$$F=\sqrt{4k_{B}TDB},$$
where *k_B_* is the Boltzmann constant, *T* is the absolute temperature, *B* is the noise bandwidth. The signal transfer function of the ball-disk rotor micro-gyro mechanical structure is given by:(3)$$S_{mech}=\frac{2I\omega R_{2}}{E}\Omega ,$$
where *ω* is the rotor rotation speed, *Ω* is the input angular velocity, *R*_2_ is the rotor ring radius. Then the signal-to-noise ratio of the micro gyroscope mechanical structure can be calculated by Equations (2) and (3).
(4)$$SNR_{mech}^{}=\frac{I\omega R_{2}\Omega }{E\sqrt{k_{B}TDB}},$$

According to this Equation (4), if the input angular velocity is given, the signal-to-noise ratio of the micro-gyro mechanical structure is proportional to the moment of inertia, rotor speed and rotor radius, and inversely proportional to the radial component coefficient of the electromagnetic force.

In order to express the relationship between the micro gyroscope structure and its corresponding noise characteristics, when the value of the input angular velocity is 1, the equivalent input angular velocity noise of the mechanical structure can be obtained:(5)$$\Omega_{mech}=\frac{E\sqrt{k_{B}TDB}}{I\omega R_{2}},$$
where *Ω_mech_* is the equivalent input angular velocity noise of mechanical structure. It can be known from Equation (5) that the equivalent input angular velocity noise of the mechanical structure is related to the damping parameters, the moment of inertia, the elastic coefficient and the rotor speed. In the practical design, the mechanical noise can be reduced by decreasing the damping and electromagnetic force of the ball disk rotor or increasing the moment of inertia and speed of the ball disk rotor.

### 2.2. Driving Noise

The driving interference in the circuit refers to the interference caused by the electromagnetic driving of the micro gyroscope to the detection part of the system. The drive method of the ball dish rotor is electromagnetic drive, and the relationship between the corresponding drive interference fundamental frequency and the micro gyroscope rotor speed is:(6)$$\omega_{0}=2\pi \frac{N}{60},$$
where *N* is the rotor rotation speed. The equivalent equation for driving interference coupled to the sense input is:(7)$$V_{n-d}^{}=A_{0}\sin\omega_{0}t+A_{1}\sin\frac{\omega_{0}}{2}t+\cdots +A_{n}\sin\frac{\omega_{0}}{n+1}t+\cdots ,$$
where *A_n_* is the amplitude of the n-th harmonic of the driving interference; *ω*_0_ is the fundamental angular frequency of the drive disturbance. It can be known from Equation (7) that the driving interference noise is mainly affected by the rotor speed and the driving force. In order to effectively reduce the driving interference noise, a high-order low-pass filter needs to be used in the detection circuit.

When the angular velocity signal is input, the micro-gyro detection capacitance will change accordingly. The relationship between the two can be expressed as:(8)$$\Delta C=\frac{IR_{2}\omega }{E}\Omega C_{0},$$
where *C*_0_ is the nominal value of the detection capacitor. Then the signal-to-noise ratio of the driving interference can be calculated by Equations (7) and (8):(9)$$SNR_{driv}^{}=\frac{IR_{2}\omega }{E}\Omega \frac{C_{0}}{A_{0}\sin\omega_{0}t},$$

It can be known from Equation (9) that the signal-to-noise ratio of driving interference is proportional to the moment of inertia, rotor speed, rotor radius, and detection capacitance. Additionally, it is inversely proportional to the radial component coefficient of the electromagnetic force and the amplitude of the driving interference signal.

In order to express the relationship between the driving interference of the micro gyroscope and its corresponding noise characteristics, when the value of the input angular velocity is 1, the equivalent input angular velocity noise of the driving interference can be obtained:(10)$$\Omega_{driv}=\frac{EA_{0}\sin\omega_{0}t}{IR_{2}\omega C_{0}},$$
where *Ω_driv_* is the driving interference equivalent input angular velocity noise. It can be known from Equation (10) that the equivalent input angular velocity noise of the drive disturbance is related to the amplitude of the drive disturbance, the damping parameter, the moment of inertia, the detection capacitance, the elastic coefficient and the rotor speed. In actual design, driving noise can be reduced by using a high-order low-pass filter in the detection circuit.

The drive interference test chart of a ball-disc rotor micro-gyro through the back electromotive force detection is shown in Figure 2. The drive interference frequency is about 0.3 KHz. At this time, the rotor speed is 18,000 rpm, and the oscilloscope sampling period is 50 KHz. Using MATLAB to process the data in Figure 2, spectral characteristics of the micro-gyro driving noise can be obtained. As shown in Figure 3, the fundamental frequency of the driving noise is 0.293 KHz, which basically matches the corresponding frequency of the rotor speed of 0.3 KHz. However, because the influence of driving noise on the output signal can be up to several hundred millivolts, the interference of driving noise must be eliminated. The effective bandwidth of the output signal is only one order of magnitude different from the driving noise. It is necessary to use higher order low-pass filter with more than six orders.

### 2.3. Electronic Noise

As shown in Figure 4, an open-loop system of multistage noise injection is proposed. According to Friis formula, the total equivalent input noise coefficient can be expressed as:(11)$$V_{in-noise}=V_{n1}+\frac{V_{n2}-1}{A_{1}}+\frac{V_{n3}-1}{A_{1}A_{2}},$$
where *V_in-noise_* is the total equivalent input noise coefficient; *V_ni_* is the noise injection of level i; *A_N_* is the gain of nth stage.

According to the relation of Equation (11), if the gain of the front stage amplifier is large enough, the sum of noise sources introduced by the rear stage can be ignored. The detection circuit of the micro gyroscope consists of the front stage transimpedance amplifier, the backstage amplifier, the modem and the low-pass filter. The main noise source of the micro gyro detection circuit can be considered to be from the front stage transimpedance amplifier, because the former stage transimpedance amplifier conforms to the previous assumption [17].

Figure 5 shows the noise model of the previous transimpedance amplifier. As shown in Figure 5, *V_rt_* denotes the high-frequency excitation noise; *C_s_* denotes the micro gyroscope sensitive structural capacitor; *V_n-__am_* denotes the input reference thermal noise of a transimpedance operational amplifier; *R_in_* denotes the input impedance of a transimpedance operational amplifier; *C_f_* denotes the feedback capacitor; *R_f_* denotes the feedback resistor; *V_r1_* denotes the feedback resistance thermal noise, *A*_0_ denotes the dc gain of the transimpedance op amp [18,19].

According to Kirchhoff’s circuit law and the characteristics of the amplifier, the equivalent input noise expression of the transimpedance amplifier can be obtained.
(12)$$\begin{array}{l} \frac{\bar{V_{n-in}^{2}}(f)}{\Delta f}=\frac{V_{n-in}^{}(f)V_{n-in}^{*}(f)}{\Delta f}=[{\frac{(2C_{s}+C_{f}}{2C_{s})}}^{2}+\frac{1+{\left(2\pi fC_{f}R_{f}\right)}^{2}}{{\left(4\pi fC_{s}R_{f}\right)}^{2}}\\ +(2C_{s}+C_{f})\frac{1+8\pi^{2}f^{2}C_{s}C_{f}R_{f}{}^{2}}{C_{s}{\left(4\pi fC_{s}R_{f}\right)}^{2}}]\times \frac{\bar{V_{n-am}^{2}}(f)}{\Delta f}+\frac{\bar{V_{Rt}^{2}}(f)}{\Delta f}+\frac{kT}{4R_{f}{\left(\pi fC_{s}\right)}^{2}} \end{array}$$

Since ${\left(2\pi fC_{f}R_{f}\right)}^{2}>>1$, $8\pi^{2}f^{2}C_{s}C_{f}R_{f}{}^{2}>>1$, and *R_f_* is very large, so Equation (12) can be further simplified as:(13)$$\frac{\bar{V_{n-in}^{2}}(f)}{\Delta f}={\left(\frac{C_{s}+C_{f}}{C_{s}}\right)}^{2}\frac{\bar{V_{n-am}^{2}}(f)}{\Delta f}+\frac{\bar{V_{Rt}^{2}}(f)}{\Delta f},$$

If *C_f_* = 2C_s_, the Equation (13) can be simplified to:(14)$$\frac{\bar{V_{n-in}^{2}}(f)}{\Delta f}=\frac{9\bar{V_{n-am}^{2}}(f)}{\Delta f}+\frac{\bar{V_{Rt}^{2}}(f)}{\Delta f},$$

From Equations (13) and (14), it can be known that the circuit noise is mainly determined by the front-stage transimpedance amplifier noise, high-frequency excitation noise, and *C_f_*/*C_s_*. The noise of the circuit can be effectively reduced by decreasing the above correlation items, but it will increase the difficulty of circuit design.

The signal-to-noise ratio of the micro gyroscope detection circuit can be calculated by Equations (8) and (14):(15)$$SNR_{cir}=\frac{I\omega R_{2}\Omega }{E}\frac{C_{0}}{{\left(\frac{C_{s}+C_{f}}{C_{s}}\right)}^{2}\bar{V_{n-am}^{2}}+\bar{V_{Rt}^{2}}},$$

According to Equation (15), when the input angular velocity is constant, the signal-to-noise ratio of the micro-gyro detection circuit is proportional to the moment of inertia, rotor speed, rotor radius, and detection capacitance; it is inversely proportional to the radial component coefficient of the electromagnetic force.

In order to express the relationship between the micro gyroscope detection circuit and its corresponding noise characteristics, when the value of the input angular velocity is 1, the equivalent input angular velocity noise of the detection circuit is defined as:(16)$$\Omega_{cir}=\frac{E[{\left(\frac{C_{s}+C_{f}}{C_{s}}\right)}^{2}\bar{V_{n-am}^{2}}+\bar{V_{Rt}^{2}}]}{I\omega R_{2}C_{0}},$$
where *Ω_cir_* is the detection circuit equivalent input angular velocity noise. It can be known from Equation (16) that the equivalent input angular velocity noise of the detection circuit is related to the previous stage transimpedance amplifier noise, high-frequency excitation noise, *C_f_*/*C_s_*, damping parameters, moment of inertia, detection capacitance, elastic coefficient, and rotor speed. In the actual design, the noise of the detection circuit can be reduced by optimizing the noise of the pre-stage transimpedance amplifier, high-frequency excitation noise, and *C_f_*/*C_s_*.

According to the different sources of noise, the noise of micro-gyroscope system can be divided into driving interference noise, mechanical noise and electronic noise. In the process of modeling, the mechanical noise caused by the thermal movement of molecules on the surface of sensitive structures, the driving interference caused by the electromagnetic driving of micro-gyroscopes and the circuit noise in the interface circuit are analyzed theoretically. As shown in Table 1, the correctness of the model is verified by the calculated values of the equivalent input angular velocity of system noise.

The equivalent input noise of the preamplifier can be calculated by dividing the output noise of the micro gyroscope system by the loop gain. The SNR of the low noise amplifier is:(17)$$SNR=\frac{V_{sig}}{V_{on}}=\frac{V_{s}(\frac{\Delta C}{C_{F}})}{\sqrt{2}V_{n-in}(1+\frac{C_{S}}{C_{F}})},$$
where *V_sig_* is the output of the preamplifier; *V_on_* is the output noise of the pre amplifier; *V_s_* is the input signal of the pre amplifier; *V_n-in_* is the equivalent input noise of the preamplifier.

According to Equation (17), the change rate of relative capacitance can be obtained as shown in Equation (18):(18)$$\frac{\Delta C}{C_{S}}=\frac{\sqrt{2}V_{n-in}(1+\frac{C_{F}}{C_{S}})}{V_{s}}=\frac{2V_{n-in}(1+\frac{C_{F}}{C_{S}})}{V_{DD}-V_{SS}},$$

From Equation (18), the minimum resolution of the system is proportional to the input noise. With the decreasing of input noise, the detection resolution of the system can be improved accordingly.

## 3. Low Noise Interface Circuits

In Figure 6, four pairs of differential detection capacitors (C_1_, C_2_, C_3_ and C_4_ are variable capacitors between the upper plate and the rotor; C_5_, C_6_, C_7_ and C_8_ are variable capacitors between the lower plate and the rotor) are used to represent the simplified equivalent model of the structure of the micro gyroscope. The driving circuit consists of OP1 and OP2 and their corresponding peripheral components, which generates sinusoidal excitation signals with equal amplitude and opposite phase. The excitation signals are loaded on the upper and lower electrodes of the gyroscope structure respectively. When the gyro rotor deflects in the X+ direction, C_1_ and C_5_ also change accordingly. The differential capacitance changes through charge-voltage conversion circuit composed of OP3, AC amplifier circuit composed of OP4, phase-sensitive demodulation circuit and LPF1 low-pass filter. The displacement output signal V_out11_ in the X+ direction is obtained. At the same time, the displacement output signal V_out1_ is amplified by the afterburner circuit K_1_ and K_5_ to form a feedback voltage-loaded differential capacitor on the control electrodes of C_1_ and C_5_, so that the rotor can return to the balanced position in the X+ direction. Similarly, changes in other directions can be obtained.

As shown in Figure 7, there is a single path rotor displacement detection circuit in *x*-axis direction, including preamplifier, AC amplifier, phase sensitive demodulation unit, low-pass filter and other modules.

The charge-voltage conversion circuit structure of the transimpedance structure is shown in Figure 8. In order to improve the open-loop gain, a three-stage folded cascode structure is used in the operational amplifier. Through the analysis of its structure, the equivalent input noise density is obtained as follows:
(19)$$V_{n}^{2}=\frac{16kT}{3}\left(\frac{1}{g_{m2}}+\frac{g_{m5}+g_{m8}}{g_{m2}^{2}}\right),$$

T-type topology is adopted to achieve the high resistance required by the trans-resistance structure. Among them, Q17 and Q19 are set to make Q18 obtain smaller gate voltage, and the transistor Q18 is designed as a long channel type, that is, smaller W/L. At this time, Q18 works in linear region, its resistance can reach Mega level, and its working characteristics can be guaranteed by the corresponding bias circuit. The equivalent resistance of T-type network is obtained as follows:(20)$$R_{eq}=R_{M}(1+\frac{R_{2}}{R_{1}})+R_{2},$$

*R_M_* is the resistance value of transistor Q18 in linear region. When the resistance of transistor Q18 is much larger than that of resistors *R*_1_ and *R*_2_ in Figure 8, Equation (20) shows that if the resistance of transistor Q18 is greater than 1 MΩ, the main noise source of charge-voltage conversion circuit is the equivalent resistance of transistor Q18. In this case, the output signal-to-noise ratio can be expressed as follows.
(21)$$SNR=\sqrt{\frac{I_{IN}^{2}R_{eq}}{4KT(R_{2}/R_{1}+1)}}\approx \sqrt{\frac{I_{IN}^{2}R_{M}}{4KT}},$$
where *I_IN_* is input signal of the charge-voltage conversion circuit. The regular expression is defined as follows.
(22)$$I_{IN}=\frac{\partial C}{\partial t}V_{b}=C_{o}V_{b}\omega \sin\omega t,$$
where *C*_0_ is maximum change value of the sensitive capacitance of the exciting electrode; *V_b_* is forward bias voltage charge voltage conversion circuit; *ω* is driving mode resonance frequency. The combination Equations (21) and (22) show that the signal-to-noise ratio is mainly determined by the resistance of transistor Q18 and the bias voltage *V_b_*.

## 4. Results and Discussion

The performance of the interface ASIC designed in this paper and the interface ASIC performance in the main literature are shown in Table 2. The relative capacitance change rates shown in the table are calculated from the measured values according to Equation (26). The vibrating micro-gyro interface ASIC designed by the University of California, Berkeley has achieved a relative capacitance change rate of 1.3 × 10^−8^, and its Sigma-Delta force feedback structure and related double sampling technology have reduced circuit noise requirements [20]. The continuous time current (CTC) sensitive circuit designed by the Technical University of Helsinki, Finland can avoid the noise coupling phenomenon of the switched capacitor circuit; the relative capacitance resolution reaches 9 × 10^−8^ [21]. The quadrature frequency-modulated micro-gyro designed by the University of California, Berkeley uses Frequency Modulation (FM) technology, and the relative capacitance resolution reaches 1.67 × 10^−8^ [22]. The Indian University of Science and Technology reported the use of bulk silicon process accelerometer switched capacitor detection circuit, whose relative capacitance resolution can reach 4.94 × 10^−9^ [23].

The layout is based on a high voltage 0.5 μm 2P2M N well CMOS process. The layout of the spherical disc rotor micro-gyroscope is shown in Figure 9. The layout area is 19.49 mm^2^. Figure 10 is the output signal noise spectrum of dynamic signal analyzer HP35670A. The test results show that the noise spectral density of interface ASIC at 10 Hz is 44.7 mV/Hz^1/2^, which is equivalent to the input noise of the system, 0.003°/s/Hz^1/2^. Test results of the interface circuit for micro gyroscope with ball-disc rotor are shown in Table 3.

As can be seen from Table 4, compared with the performance parameters of the rotor gyroscope published in recent years, the ball-disc rotor micro-gyroscope designed in this paper has low noise and high linearity, and can obtain more accurate angular velocity measurements in some occasions with high accuracy requirements.

## 5. Conclusions

In this work, we describe a low noise interface ASIC for micro gyroscope with ball-disc rotor. The transimpedance pre-amplifier of the proposed circuit effectively addresses thermal and flicker noise issues. The measurement results for the circuit implemented in a 0.5 µm CMOS technology show an equivalent to the input noise of the system, 0.003°/s/Hz^1/2^ and a sensitivity of 0.003°/s with ±100°/s measure range.

## Figures and Tables

**Figure 1 sensors-20-01238-f001:**
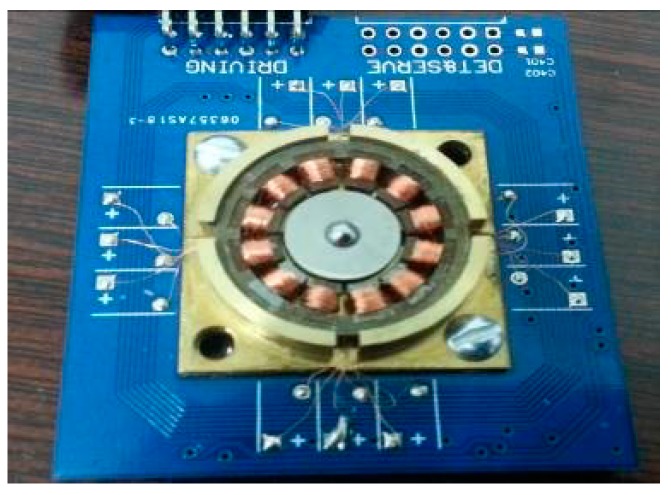
Mechanical structure of micro gyroscope with ball-disc rotor.

**Figure 2 sensors-20-01238-f002:**
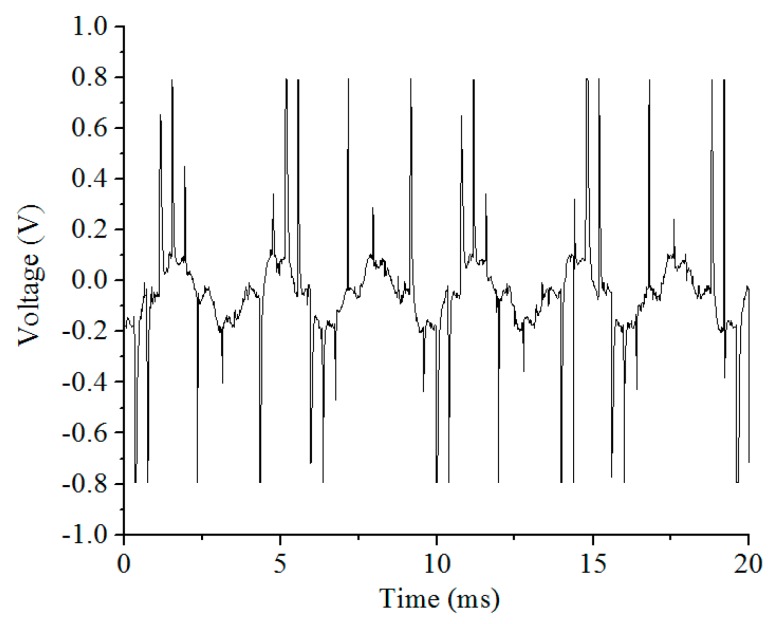
Test of driving interference for micro gyroscope with ball-disc rotor.

**Figure 3 sensors-20-01238-f003:**
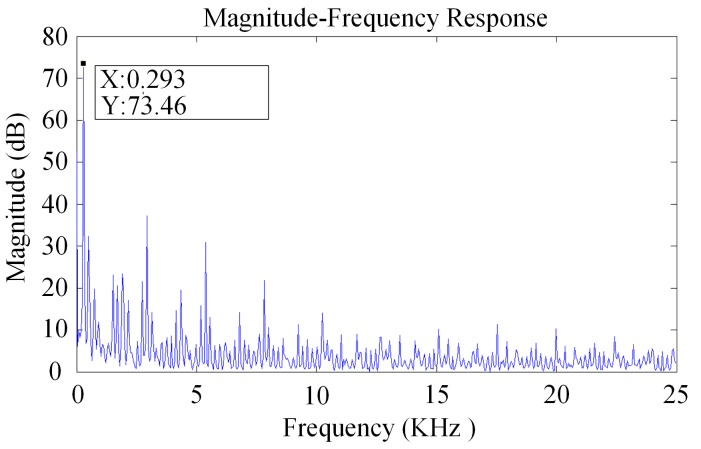
Frequency response of driving interference for micro gyroscope.

**Figure 4 sensors-20-01238-f004:**
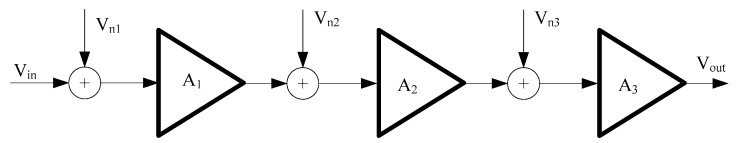
Noise of opened-loop system.

**Figure 5 sensors-20-01238-f005:**
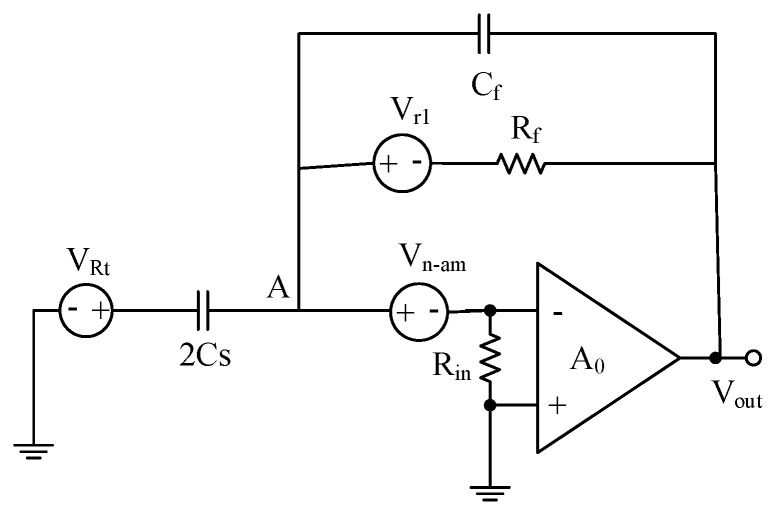
Noise model of the transimpedance amplifier.

**Figure 6 sensors-20-01238-f006:**
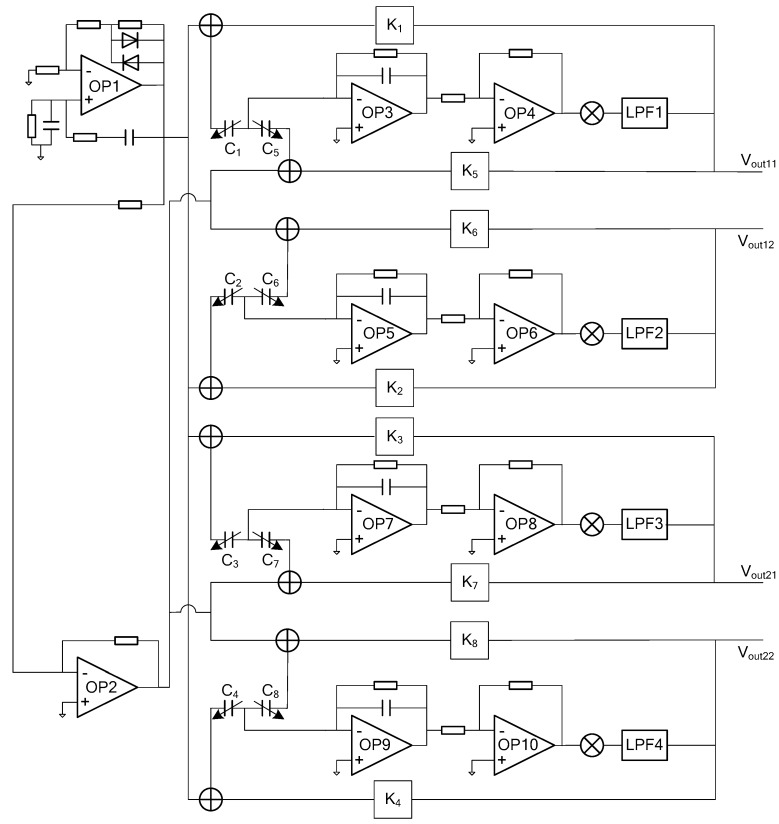
Overall realization scheme for the interface ASIC of micro gyroscope with ball-disc rotor.

**Figure 7 sensors-20-01238-f007:**
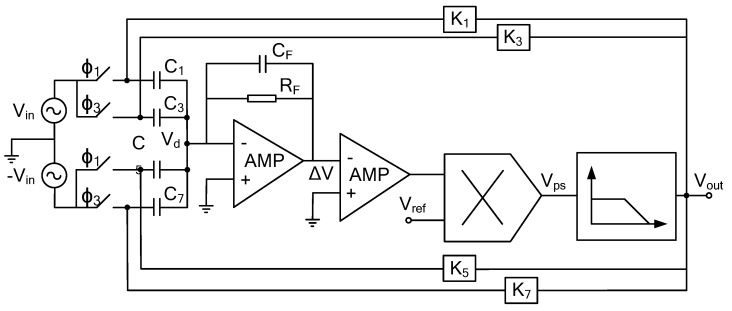
Schematic of the single path detection principle of micro gyroscope with ball-disc rotor.

**Figure 8 sensors-20-01238-f008:**
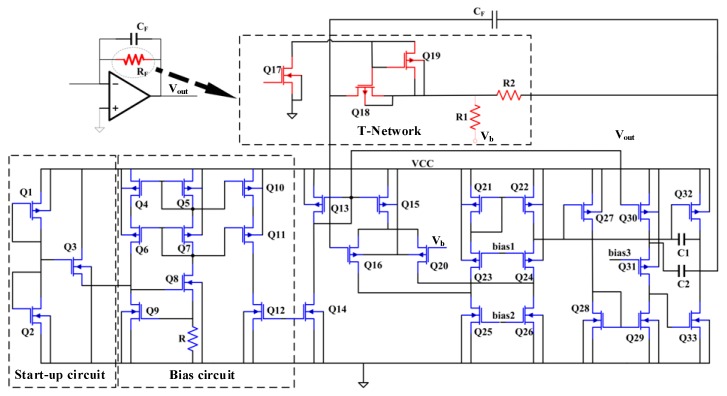
Transimpedance structure of charge voltage conversion circuit.

**Figure 9 sensors-20-01238-f009:**
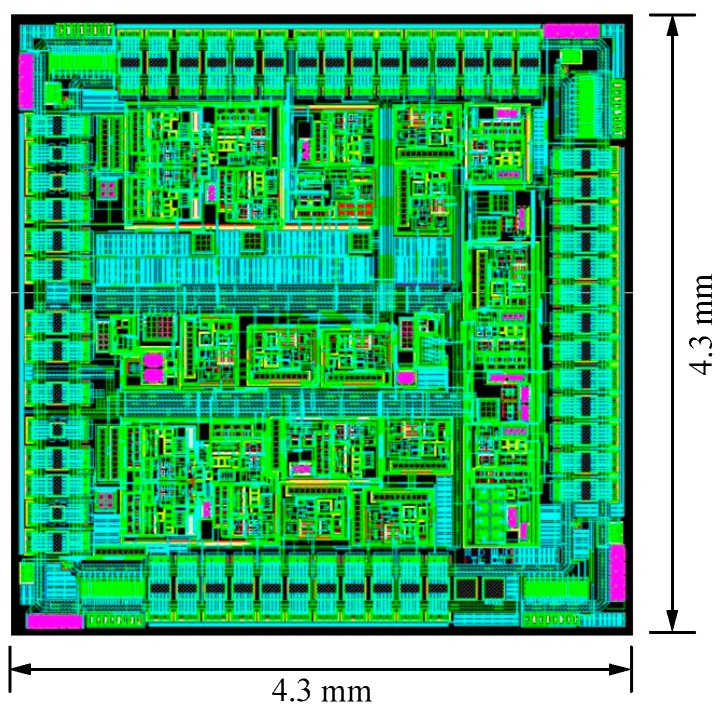
Layout of the interface circuit for micro gyroscope with ball-disc rotor.

**Figure 10 sensors-20-01238-f010:**
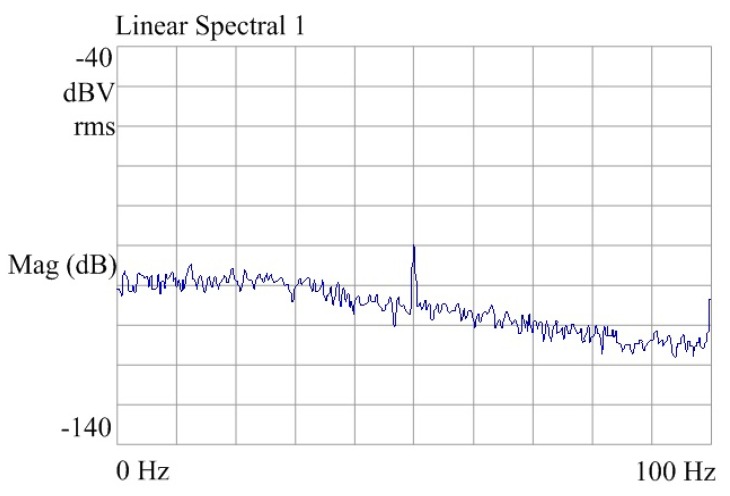
Spectrum of system output noise.

**Table 1 sensors-20-01238-t001:** Calculated values of equivalent input angular velocity noise of gyroscope system.

Micro Gyroscope Noise Source	Equivalent Input Angular Velocity	Calculated Value (°/s/Hz^1/2^)
Structural Thermal Sensitive Noise	$\Omega_{mech}=\frac{E\sqrt{k_{B}TDB}}{IR_{2}\omega }$	6.1 × 10^−4^
Driving Interference Noise	$\Omega_{driv}=\frac{EA_{0}\sin\omega_{0}t}{IR_{2}\omega C_{0}}$	4.6 × 10^−3^
Detecting Circuit Noise	$\Omega_{cir}=\frac{E[{\left(\frac{C_{s}+C_{f}}{C_{s}}\right)}^{2}\bar{V_{n-am}^{2}}+\bar{V_{Rt}^{2}}]}{IR_{2}\omega C_{0}}$	1.8 × 10^−5^

**Table 2 sensors-20-01238-t002:** Comparison of reported works of interface ASIC with this work.

	Sensor Type	Circuit Structure	Static Capacitance (pF)	Sensitivity	Noise Density	Resolution (ΔC_MIN_/CS)
[19]	gyroscope	SC	—	—	0.004°/s/Hz^1/2^	1.3 × 10^−8^
[20]	accelerometer	CTC	12	3.8 pF/g	0.3 µg/Hz^1/2^	9 × 10^−8^
[21]	gyroscope	FM	6 × 10^−3^	—	0.09°/s/Hz^1/2^	1.67 × 10^−8^
[22]	accelerometer	SC	0.88	4.576 fF/g	—	4.94 × 10^−9^
This paper	gyroscope	CTC	5	0.003°/s	0.003°/s/Hz^1/2^	1 × 10^−8^

**Table 3 sensors-20-01238-t003:** Test results of the interface circuit for micro gyroscope with ball-disc rotor.

Parameters	Measurements
Supply voltage (V)	±9
Area (mm^2^)	4.9 × 4.9
Power dissipation (mW)	30
Flicker noise corner (kHz)	100
Output voltage range (V)	–8.6–+8.3
Non-linear (%)	0.02
System output noise at 10 Hz (mV/Hz^1/2^)	44.7

**Table 4 sensors-20-01238-t004:** Comparison of reported works with this work.

Parameters	[24]	[25]	This Paper
Maximum Rotor Speed (rpm)	58,000	12,000	20,000
Nonlinear Error (%)	0.1%	0.58%	0.05%
Noise Density (°/s/Hz^1/2^)	0.03	0.015	0.003
Measure Range (o/s)	±200	±100	±100
Bandwidth (Hz)	10	10	10
Sensitivity (°/s)	0.1	0.014	0.003
Scale Factor (mV/°/s)	-	39.8	15

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
