# Peer review of "Low Noise Interface ASIC of Micro Gyroscope with Ball-disc Rotor"

_sensors, 2020, doi:10.3390/s20041238_

Round 1
Reviewer 1 Report
This paper reports the design and fabrication of a low noise ASIC interface of a microgyroscope. In general, the text is well written and the methodology clearly explained. There are only few specific issues, which have to be addressed:
1) In the introduction, lines 27-28 need revision.
2) There is some expression missing at the end of line 44: is equation (1) correctly displayed?
3) At page 7, lines 163-165, the text reports that “As shown in Table 1, the correctness of the model is verified by the theoretical and experimental results of the equivalent input angular velocity of system noise”. Additional details about the model validation should be provided, since from Table 1, it is only possible to read the “calculated values of equivalent input angular velocity noise of gyroscope system”.
4) The discussion of Table 2 should be enriched with some critical comments about the performance of the present design compared to previous (and cited) works. At the moment, there is only a list of the resolution values of previous designs.
Author Response
This paper reports the design and fabrication of a low noise ASIC interface of a microgyroscope. In general, the text is well written and the methodology clearly explained. There are only few specific issues, which have to be addressed:
- In the introduction, lines 27-28 need revision.
Answer:
Yes, I have rewritten the contents of line 1-28. The specific content is as follows
“Micro gyroscopes are small, inexpensive sensors that can measure angular rate or angle, which have increasingly taken the place of conventional rate sensors in many applications such as defense and consumer electronics [1]. Vibratory gyroscopes are the most common of the existing micro gyroscopes [2]. But it is difficult to reach tactical-grade and inertialgrade performance with these kinds of gyroscopes because of the structural constraint. Rotor gyroscope potentially has a higher performance than vibratory gyroscopes [3-5].
Noise isn’t only an area of science and technology that poses practical problems but also has deep intellectual attractions [1]. The noise in micro gyroscopes includes both external noise and internal noise. The magnitude of external noise signals coupled into a gyroscope varies with the actual conditions. Mechanical thermal noise and circuit noise are the most important sources of internal noise in micro gyroscopes. Effectively reducing internal noise and improving the signal-to-noise ratio of the system are important challenge for gyroscope designers.
The mechanical noise exists widely in micro gyroscope structures, namely Brownian motion, and is caused by random impacts of molecules on a structure, it is commonly considered as the mechanical sensitivity limitation [6-9]. And the circuit noise includes electrical-thermal noise (Johnson noise), shot noise, flicker noise (1/f noise), demodulation phase noise, and so on. To reduce this noise, low-noise readout circuits have been reported [10-16]. But, the published papers mostly focus on vibratory micro gyroscope, and the noise analysis of rotor-type micro gyroscope is rarely mentioned. Because the noise level affects the detection accuracy of micro gyroscope, it is urgent to carry out the research on the related theory and model of rotor micro gyroscope noise.”
2) There is some expression missing at the end of line 44: is equation (1) correctly displayed?
Answer:
The equation (1) should be at the end of line 44. Equation (1) was placed at line 49 due to typographical errors. Now I have adjusted the position of equation (1).
3) At page 7, lines 163-165, the text reports that “As shown in Table 1, the correctness of the model is verified by the theoretical and experimental results of the equivalent input angular velocity of system noise”. Additional details about the model validation should be provided, since from Table 1, it is only possible to read the “calculated values of equivalent input angular velocity noise of gyroscope system”.
Answer:
Yes, I have modified this text. The specific content is as follows.
“As shown in Table1, the correctness of the model is verified by the calculated values of the equivalent input angular velocity of system noise”.
4) The discussion of Table 2 should be enriched with some critical comments about the performance of At the moment, there is only a list of the resolution values of previous designs.
Answer:
Yes, I have modified Table 2. I added Sensitivity and Noise Density of the present design compared to previous (and cited) works. The specific content is as follows.
Table 2. Comparison of reported works of interface ASIC with this work.
|
Sensor type |
Circuit structure |
Static capacitance (pF) |
Sensitivity |
Noise Density |
Resolution (ΔCMIN/CS) |
[19] |
gyroscope |
SC |
— |
— |
0.004 o/s/Hz1/2 |
1.3×10-8 |
[20] |
accelerometer |
CTC |
12 |
3.8pF/g |
0.3µg/Hz1/2 |
9×10-8 |
[21] |
gyroscope |
FM |
6×10-3 |
— |
0.09 o/s/Hz1/2 |
1.67×10-8 |
[22] |
accelerometer |
SC |
0.88 |
4.576 fF/g |
— |
4.94×10-9 |
This paper |
gyroscope |
CTC |
5 |
0.003 o/s |
0.003 o/s/Hz1/2 |
1×10-8 |

Reviewer 2 Report
The paper introduced a low noise interface ASIC of the micro gyroscope with the ball‐disc rotor. Even though the general idea of the paper seems interesting, it requires significant improvement to be considered for publication. The quality of the writing is relatively poor from word choice, weird sentence breakdowns, and most importantly wrong grammatical structures. For instance, in line 28 the authors used "however" incorrectly.
The authors were not through regarding the literature review. There are several techniques that researchers used for reducing the effect of noise and increase the signal-to-noise ratio and unfortunately, there are mostly ignored in the introduction. In my opinion, the authors should re-write the introduction by doing a complete literature review and motivate the problem in hand from what has been done in the field.
I regret to say that the paper is poorly written missing crucial details for justifying important assumptions and expressions and filled with a lot of unnecessary steps. For instance, I think they can summarize page 6 in probably one paragraph.
Even though the idea of the paper is interesting, it is hard to follow and the results are poorly presented.
Author Response
The paper introduced a low noise interface ASIC of the micro gyroscope with the ball‐disc rotor. Even though the general idea of the paper seems interesting, it requires significant improvement to be considered for publication. The quality of the writing is relatively poor from word choice, weird sentence breakdowns, and most importantly wrong grammatical structures. For instance, in line 28 the authors used "however" incorrectly.
Answer:
Yes, I have rewritten the contents. The specific content is as follows
“The mechanical noise exists widely in micro gyroscope structures, namely Brownian motion, and is caused by random impacts of molecules on a structure, it is commonly considered as the mechanical sensitivity limitation [6-9]. And the circuit noise includes electrical-thermal noise (Johnson noise), shot noise, flicker noise (1/f noise), demodulation phase noise, and so on. To reduce this noise, low-noise readout circuits have been reported [10-16]. But, the published papers mostly focus on vibratory micro gyroscope, and the noise analysis of rotor-type micro gyroscope is rarely mentioned. Because the noise level affects the detection accuracy of micro gyroscope, it is urgent to carry out the research on the related theory and model of rotor micro gyroscope noise.”
The authors were not through regarding the literature review. There are several techniques that researchers used for reducing the effect of noise and increase the signal-to-noise ratio and unfortunately, there are mostly ignored in the introduction. In my opinion, the authors should re-write the introduction by doing a complete literature review and motivate the problem in hand from what has been done in the field.
Answer:
Yes, I have rewritten the contents. The specific content is as follows
“Micro gyroscopes are small, inexpensive sensors that can measure angular rate or angle, which have increasingly taken the place of conventional rate sensors in many applications such as defense and consumer electronics [1]. Vibratory gyroscopes are the most common of the existing micro gyroscopes [2]. But it is difficult to reach tactical-grade and inertialgrade performance with these kinds of gyroscopes because of the structural constraint. Rotor gyroscope potentially has a higher performance than vibratory gyroscopes [3-5].
Noise isn’t only an area of science and technology that poses practical problems but also has deep intellectual attractions [1]. The noise in micro gyroscopes includes both external noise and internal noise. The magnitude of external noise signals coupled into a gyroscope varies with the actual conditions. Mechanical thermal noise and circuit noise are the most important sources of internal noise in micro gyroscopes. Effectively reducing internal noise and improving the signal-to-noise ratio of the system are important challenge for gyroscope designers.
The mechanical noise exists widely in micro gyroscope structures, namely Brownian motion, and is caused by random impacts of molecules on a structure, it is commonly considered as the mechanical sensitivity limitation [6-9]. And the circuit noise includes electrical-thermal noise (Johnson noise), shot noise, flicker noise (1/f noise), demodulation phase noise, and so on. To reduce this noise, low-noise readout circuits have been reported [10-16]. But, the published papers mostly focus on vibratory micro gyroscope, and the noise analysis of rotor-type micro gyroscope is rarely mentioned. Because the noise level affects the detection accuracy of micro gyroscope, it is urgent to carry out the research on the related theory and model of rotor micro gyroscope noise.”
I regret to say that the paper is poorly written missing crucial details for justifying important assumptions and expressions and filled with a lot of unnecessary steps. For instance, I think they can summarize page 6 in probably one paragraph.
Answer:
Yes, I have summarized page 6. The specific content is as follows
“According to Kirchhoff's circuit law and the characteristics of the amplifier, the equivalent input noise expression of the transimpedance amplifier can be obtained.
, |
(21) |
Since, , and Rf is very large, so equation (21) can be further simplified as:
, |
(22) |
If Cf=2Cs,the equation (22) can be simplified to:
, |
(23) |
From equations (21) and (22), it can be known that the circuit noise is mainly determined by the front-stage transimpedance amplifier noise, high-frequency excitation noise, and Cf/Cs. The noise of the circuit can be effectively reduced by decreasing the above correlation items, but it will increase the difficulty of circuit design.”
Even though the idea of the paper is interesting, it is hard to follow and the results are poorly presented.
Answer:
Yes, I have modified data of Table 2, and added Table 3. I added Sensitivity and Noise Density of the present design compared to previous (and cited) works. Table 3 is the test results of the interface circuit for micro gyroscope with ball-disc rotor. The specific content is as follows.
Table 2. Comparison of reported works of interface ASIC with this work.
|
Sensor type |
Circuit structure |
Static capacitance (pF) |
Sensitivity |
Noise Density |
Resolution (ΔCMIN/CS) |
[19] |
gyroscope |
SC |
— |
— |
0.004 o/s/Hz1/2 |
1.3×10-8 |
[20] |
accelerometer |
CTC |
12 |
3.8pF/g |
0.3µg/Hz1/2 |
9×10-8 |
[21] |
gyroscope |
FM |
6×10-3 |
— |
0.09 o/s/Hz1/2 |
1.67×10-8 |
[22] |
accelerometer |
SC |
0.88 |
4.576 fF/g |
— |
4.94×10-9 |
This paper |
gyroscope |
CTC |
5 |
0.003 o/s |
0.003 o/s/Hz1/2 |
1×10-8 |
Table 3. Test result of the interface circuit for micro gyroscope with ball-disc rotor
Parameters |
Measurements |
Supply voltage(V) |
±9 |
Area(mm2) |
4.9 × 4.9 |
Power dissipation(mW) |
30 |
Flicker noise corner(kHz) |
100 |
Output voltage range (V) |
-8.6 - 8.3 |
Non-linear(%) |
0.02 |
System output noise at 10 Hz(mV/Hz1/2) |
44.7 |

Round 2
Reviewer 2 Report
I think the paper is improved, but there is still room for improvement in presenting the results or in written English. I do acknowledge the fact that the authors are not native English speakers, but I believe the flow of the paper can be improved.
This manuscript is a resubmission of an earlier submission. The following is a list of the peer review reports and author responses from that submission.